# Antithetic hTERT Regulation by Androgens in Prostate Cancer Cells: hTERT Inhibition Is Mediated by the ING1 and ING2 Tumor Suppressors

**DOI:** 10.3390/cancers13164025

**Published:** 2021-08-10

**Authors:** Sophie Bartsch, Kimia Mirzakhani, Laura Neubert, Alexander Stenzel, Marzieh Ehsani, Mohsen Esmaeili, Thanakorn Pungsrinont, Merve Kacal, Seyed Mohammad Mahdi Rasa, Julia Kallenbach, Divya Damodaran, Federico Ribaudo, Marc-Oliver Grimm, Francesco Neri, Aria Baniahmad

**Affiliations:** 1Institute of Human Genetics, Jena University Hospital, 07740 Jena, Germany; sophie.bartsch@gmx.de (S.B.); kimia.mirzakhani@uni-jena.de (K.M.); laura.neubert@leibniz-fli.de (L.N.); alstenzel@ukaachen.de (A.S.); marzieh.ehsani@med.uni-jena.de (M.E.); Mohsen.esmaeili@sickkids.ca (M.E.); thanakorn.pungsrinont@med.uni-jena.de (T.P.); merve.kacal@ki.se (M.K.); julia.kallenbach@uni-jena.de (J.K.); Damodaran.DivyaLakshmi@sysmex-rdce.com (D.D.); federico.ribaudo@helmholtz-muenchen.de (F.R.); 2Leibniz Institute on Aging, 07745 Jena, Germany; mahdi.rasa@leibniz-fli.de (S.M.M.R.); francesco.neri@leibniz-fli.de (F.N.); 3Department of Adult and Pediatric Urology, Jena University Hospital, 07747 Jena, Germany; marc-oliver.grimm@med.uni-jena.de

**Keywords:** ING1, ING2, tumor suppressor, androgen receptor, prostate cancer, telomerase expression

## Abstract

**Simple Summary:**

The expression of the catalytic subunit of the human telomerase reverse transcriptase subunit (hTERT) is hormonally controlled. Androgen treatment suppresses the hTERT expression at a transcriptional level in prostate cancer cells. Here, we identified the responsive promoter element that mediates the androgen receptor induced transrepression of *hTERT*. The negative androgen response element (nARE) is identified as 62 bp located in the core promoter of *hTERT*. Chromatin immunoprecipitations indicate an androgen-dependent recruitment of the androgen receptor (AR) ING1 and ING2 to the *hTERT* promoter. Interestingly, the androgen-induced transrepression is mediated by the class II tumor suppressors inhibitor of growth 1 and 2, namely ING1 and ING2, respectively.

**Abstract:**

The human telomerase is a key factor during tumorigenesis in prostate cancer (PCa). The androgen receptor (AR) is a key drug target controlling PCa growth and regulates *hTERT* expression, but is described to either inhibit or to activate. Here, we reveal that androgens repress and activate *hTERT* expression in a concentration-dependent manner. Physiological low androgen levels activate, while, notably, supraphysiological androgen levels (SAL), used in bipolar androgen therapy (BAT), repress *hTERT* expression. We confirmed the SAL-mediated gene repression of *hTERT* in PCa cell lines, native human PCa samples derived from patients treated ex vivo, as well as in cancer spheroids derived from androgen-dependent or castration resistant PCa (CRPC) cells. Interestingly, chromatin immuno-precipitation (ChIP) combined with functional assays revealed a positive (pARE) and a negative androgen response element (nARE). The nARE was narrowed down to 63 bp in the *hTERT* core promoter region. AR and tumor suppressors, inhibitor of growth 1 and 2 (ING1 and ING2, respectively), are androgen-dependently recruited. Mechanistically, knockdown indicates that ING1 and ING2 mediate AR-regulated transrepression. Thus, our data suggest an oppositional, biphasic function of AR to control the *hTERT* expression, while the inhibition of *hTERT* by androgens is mediated by the AR co-repressors ING1 and ING2.

## 1. Introduction

Prostate cancer (PCa) is the most commonly diagnosed cancer and the second leading cause of cancer mortality for males in the USA [1]. The androgen receptor (AR) regulates the development and physiological function of the normal prostate, as well as the proliferation of cancerous prostate tissue. PCa is initially an androgen-sensitively growing tumor that eventually becomes castration-resistant (CRPC) and drug resistant [2]. While age is a major risk factor for this disease, androgen levels also contribute to the growth of the tumor. Importantly, AR represents a major drug target in the treatment of PCa. An important hormone therapy is androgen deprivation therapy (ADT) to reduce androgen production. This is often combined with AR-antagonist treatment for full blockade of the AR.

Interestingly however, there is a paradox in the androgen response of PCa cells. In addition to androgen deprivation, supraphysiological levels of androgens (SAL) also inhibit PCa growth [3,4,5]. This observation led to clinical trials using bipolar androgen therapy (BAT) composed of applying SAL within repeated cycles in combination with androgen deprivation [6]. Supraphysiologic levels of testosterone used for BAT have been shown to be effective for inhibiting PCa progression. It is suggested that the sudden switch in the androgen levels provide less time for adaptive responses of AR-signaling [7]. Mechanistically, we found that SAL induces cellular senescence in human PCa tumor cell lines, as well as in PCa tissues obtained from patients with prostatectomy [8]. This suggests that SAL may induce a tumor suppressive program in PCa cells. In line with this, SAL represses *hTERT* expression, the catalytical subunit of the telomerase [9], which might be a factor as a tumor-suppressive function of AR.

AR is a ligand-controlled transcription factor and a member of the steroid and nuclear hormone receptor superfamily. Androgen-activated AR translocates to the nucleus, where it binds to chromatin at androgen response elements (ARE) to control the gene expression. AR can interact with co-activators and co-repressors to either activate or repress the expression of target genes [10,11].

The regulation of *hTERT* expression by androgen is controversial. On the one hand, the induction of *hTERT* by androgens was reported [12]. Mechanistically, an indirect effect of androgens on *hTERT* activation was postulated [13]. The use of AR antagonists indicate a down-regulation of *hTERT* leading also to the assumption that androgens activate *hTERT* [14]. However, on the other hand, the first evidence of androgen regulation of telomerase activity was obtained from castrated animals [15]. In line with the role of androgens as a protective agent, androgen ablation in animal model systems led to the induction of telomerase activity [15,16]. These data suggest that the telomerase activity is repressed by androgens, and reducing the androgen levels leads to its activation. The underlying mechanism of how AR activates or represses *hTERT* expression remains elusive. It has further been reported that *hTERT* is repressed by androgen treatment in PCa cells [9,12,17].

As we have recently identified novel AR co-repressors that act as tumor suppressors [18,19], we aimed to investigate the underlying mechanism of *hTERT* repression by androgens. The inhibitor of growth (ING) family of proteins represents a tumor suppressor family [20]. Unlike the other ING family members, human ING1 and ING2 are closely related proteins exhibiting a high homology in the amino acid sequence [21,22]. Interestingly, ING1 and ING2 have been found to physically interact with AR and to act as AR co-repressors [18,19]. It remains unclear whether ING1 and ING2 are also involved in AR-mediated transrepression. We also included an analysis of the first- and second-generation AR-antagonists Bicalutamide, Enzalutamide (Enz), and Darolutamide.

Here, we show that both the androgen-mediated activation and inhibition of *hTERT* expression are androgen dose-dependent. Furthermore, the data suggest that the androgen-regulation of *hTERT* expression is time-dependent. In native human PCa patient samples treated ex vivo, the expression of *hTERT* is repressed by SAL. The *hTERT* repression is profound in SAL, whereas the physiological low androgen levels (LAL) instead activate the *hTERT* expression. In line with this, we identified in the distal *hTERT* promoter a positive androgen response element and in the proximal promoter region a negative androgen response element, nARE, spanning 63 bp. Interestingly, the androgen-regulated *hTERT* repression was mediated through ING1 and ING2. Notably, AR antagonists also repressed the *hTERT* expression, albeit less efficiently than SAL.

## 2. Materials and Methods

### 2.1. Cell Culture and Retroviral Transduction

The cultivation of both LNCaP and C4-2 cells were described previously [8]. As defined earlier based on androgen-dependent growth, we used 1nM R1881 or 10nM DHT as SAL, 1pM R1881 or 10pM DHT as LAL, and DMSO as the solvent control [8]. For the knockdown of ING1 and ING2, retroviral vectors pLMP-shING1b and pSR-shING2a were used with pLMP-shluc or pSR-shluc as the controls, respectively. Transient transfection and retroviral transduction experiments were performed as described earlier [9,18,19]. Transient transfections were performed with the indicated *hTERT* promoter fragments fused to the luciferase gene. The *hTERT*-3996 fragment harbors the entire promoter sequence fused to the luciferase gene, while the sequence between the numbers indicates the *hTERT* promoter regions fused to the TATA-Box-luciferase gene fusion.

### 2.2. RNA-Sequencing and Transcriptome Analysis

The total RNA was isolated from LNCaP and C4-2 cells treated for 72 h with solvent control and androgens, in triplicate, using peqGOLD TriFast (Peqlab, Erlangen, Germany), according to the manufacturer’s protocol. The sequencing library was prepared using SMARTer Stranded Total RNA Sample Prep Kit—HI Mammalian (TAKARA, Kusatu, Japan). The paired-end sequencing was done using Illumina NextSeq 500/550 High-Output v2 Kit (150 cycles, Illumina, San Diego, CA, USA). Fastq files’ quality check was performed using FastQC (v0.11.5). The first four nucleotides were removed from the sequenced reads using fastx_trimmer (FASTX Toolkit 0.0.13). The low-quality nucleotides from the end of each read were removed using fastq_quality_trimmer (FASTX Toolkit 0.0.13) with -Q 33 -t 20 -l 25 parameters. Only the first sequenced read from the pair end (R1) was used for the downstream analysis. The fastq files were mapped to the hg19 genome using tophat (v2.1.0) with the following parameters --bowtie1 --no-coverage-search -a 5. The number of reads covered by each gene was calculated by htseq-count (0.11.2) with -s no -a 0 -t exon -m intersection-nonempty parameters and hg19 gencode.v19 annotation. Before further analysis, all of the rRNA genes (5srRNA, rRNA, and mt-rRNA) were removed from the count data. To calculate the *p*-value and normalized count (based on the geometric library size factors), DESeq2 (1.20.0) R package with the default parameters and paired test was used. For the gene ontology analysis, the deferentially expressed genes were used in gprofiler function from gProfileR (v0.7.0) R package, searching in the GO:BP, GO:MF, GO:CC, and REAC data bases.

For the gene set enrichment analysis (GSEA), normalized counts (for each gene in all of the samples) were scaled using the scale function in R (with center = TRUE, scale = TRUE parameters). The average of the z-scores (scaled normalized counts) was calculated for each group (one value for each gene per group) and was used for plotting and statistical analysis. Wilcoxon paired test was performed for *p*-value calculation. Reactome analysis was performed according to [23]. The RNA-sequencing data are available in the gene expression omnibus (GEO) database under the accession number GSE151492.

### 2.3. Generation of Spheroids and Immunostaining

Spheroids were generated, as described earlier [24], by forced floating methods in 96-well ULA plates with seeding of 1000 cells/well. Then, 24 h after seeding, AR ligands or DMSO were added with each six technical replicates.

### 2.4. Senescence-Associated Beta-Galactosidase (SA β-Gal) Staining of Spheroids

After eight days of treatment, the spheroids were fixed in 4% paraformaldehyde and washed three times with 1× PBS pH 6.0. For the SA β-Gal activity staining, the spheroids were incubated in the staining solution according to [8].

### 2.5. Reporter Gene Assay

The reporter gene assays were performed as described earlier [9,18,19].

### 2.6. Reverse Transcription Quantitative Real-Time PCR (RT-qPCR)

RNA was isolated from the cells using peqGOLD TriFast (Peqlab, Erlangen, Germany), according to the manufacturer’s protocol. Two-step RT-qPCR was conducted using a High Capacity cDNA Reverse Transcription Kit (Applied Biosystems, Foster City, CA, USA) and the SsoFast EvaGreen Supermix (Bio-Rad, Munich, Germany), gene specific primers, and Bio-Rad CFX96 Real Time PCR detection system. The RT-qPCR results were analyzed via the ΔΔCt method [25] using CFX manager software from Bio-Rad. The primer sequences are listed as 5′→3′:
*β-Actin:*fwd:CACCACACCTTCTACAATGAGC
rev:CACAGCCTGGATAGCAACG*RPL13a:*fwd:GTATGCTGCCCCACAAAACC
rev:TGTAGGCTTCAGACGCACGAC*TBP:*fwd:GGCGTGTGAAGATAACCCAAGG
rev:CGCTGGAACTCGTCTCACT*KLK3:*fwd:GAGGCTGGGAGTGCGAGAAG
rev:TTGTTCCTGATGCAGTGGGC*hTERT:*fwd:CGGAAGAGTGTCTGGAGCAA
rev:GGATGAAGCGGAGTCTGGA

### 2.7. Antibodies and Western Blot Analyses

Preparation of whole cell lysates and Western blotting were performed as described elsewhere [18]. The primary antibodies used for immunodetection were ING1 (BD Biosciences, 550455, Franklin Lakes, NJ, USA), ING2 (Proteintech, 11560-1-AP, Manchester, UK), hTERT (Merck Chemicals, ABE2075, Darmstadt, Germany), β-Actin (Abcam, ab6276, Cambridge, UK), and α-Tubulin (Abcam, ab15246). As secondary antibodies, horseradish peroxidase-conjugated anti-mouse IgG (Santa Cruz, sc-2005, Dallas, TX, USA) or anti-rabbit IgG (Santa Cruz, sc-2370) were used. LabImage 1D software (Kapelan Bio Imaging solutions, Leipzig, Germany) was applied for the quantification of the proteins of interest relative to the loading control (α-Tubulin or β-Actin).

### 2.8. Ex Vivo Treatment of Prostate Cancer Samples

The ex vivo treatment of native PCa samples from patients with radical prostatectomies was described previously [26]. All of the patients gave informed consent and all were informed about the purpose of the study. The study was approved by the Ethics Committee of the Friedrich Schiller University (ethical approvals 3286–11/11 and 2019-1502), and it conformed to the Declaration of Helsinki.

### 2.9. Chromatin Immunoprecipitation (ChIP)

ChIP was essentially performed as described previously [9], with the modification that the beads were washed with RIPA buffer and suspended in a 100 µL RIPA buffer. The antibodies were used according to Table 1. For the *hTERT* genomic proximal promoter locus, nested PCR was used.

### 2.10. Primer Pairs for Human Genes Used for qPCR Analysis

Primer pairs for indicated target genes with sequence 5′→3′.

### 2.11. KLK3, PSA ARE I

Fwd: TCTGCCTTTGTCCCCTAGAT

Rev: AACCTTCATTCCCCAGGACT

### 2.12. Primer Pairs for hTERT Used for Nested-PCR Analysis with Sequence 5′→3′

*TERT*, Proximal promoter first primer set

Fwd: CTCCCAGTGGATTCGCGG

Rev: CGGGGCCAGGGCTTC

*TERT*, second primer set

Fwd: TGCCCCTTCACCTTCCAG

Rev: GCGCTGCCTGAAACTCGC

### 2.13. Statistical Analysis

For the statistical analyses, except otherwise stated, two-tailed unpaired Student’s t-tests were performed using the GraphPad Prism 8.0. Within each experiment, treatments were performed in technical duplicates. For the statistical analysis, data from three independent experiments were analyzed. Western Blot analysis and RT-qPCR experiments were performed for at least two independent experiments. Error bars show the mean ± SEM values, calculated from the mean, SD, and number of independent replicates. For statistical analysis of spheroid growth repeated measure two-way ANOVA was performed.

A 95% confidence interval (*p*-value, *p* < 0.05) was considered as statistically significant (*) between two subject groups. A 99% confidence interval (*p* < 0.01) and a 99.9% confidence interval (*p* < 0.001) were indicated by two (**) and three stars (***), respectively.

## 3. Results

### 3.1. Molecular Profiling on Transcriptomics Level Reveals Repression of hTERT Expression by SAL

To identify the mRNA profile affected by androgen treatment at SAL, we performed transcriptome analyses with multiple comparisons. RNA sequencing (RNA-Seq) was performed with both the androgen-sensitive human LNCaP and the CRPC C4-2 cell lines treated with and without the androgen R1881 (methyltrienolone) at SAL (1 nM) or with DMSO as the solvent control. As dihydrotestosterone (DHT) can be rapidly metabolized and the metabolites can activate the estrogen receptor beta [27], the more stable and AR-specific compound R1881 was used as an AR agonist to treat the cells for 72 h. RNA-seq has been performed three times independently for each condition in both cell lines with all treatments. The number of significantly differentially expressed genes (DEGs), including the number of upregulated genes and the number of downregulated genes, indicates that LNCaP responds more sensitively to androgen treatment compared with the CRPC cell line C4-2 (Figure 1A). This may be due to the fact that the AR is more active in CRPC cells, and thus additional added ligands lead to a lower number of DEGs compared with the LNCaP cells.

Significantly down-regulated genes after androgen treatment were significantly enriched in telomere-related gene ontology (GO) terms in both the LNCaP and C4-2 cell lines (Figure 1B,D). Interestingly, upon androgen treatment at SAL, enrichment of the down-regulated genes was observed in terms such as telomere organization, telomere capping, and telomerase activity in both cell lines (Figure 1B,D). Furthermore, the reactome terms indicate that the significantly down-regulated genes were also enriched in telomeric terms by SAL treatment in both cell lines (Figure 1C,E). No significantly enriched GO term was observed from the up-regulated genes in both cell lines. The number of enriched terms in LNCaP was higher compared with the C4-2 cells, indicating that telomeric function is less androgen-regulated in CPPC C4-2 cells, and telomere-related genes in LNCaP show a stronger response to androgen treatment (Figure 1F). This observation was confirmed by analyzing the shelterin complex gene set (Figure 1F). The shelterin complex is a multi-subunit protein complex responsible for protecting telomeric DNA from unwanted degradation and end-to-end fusion events [28]. These data suggest an androgen regulation of telomere protecting factors more significantly in LNCaP compared with C4-2 cells (Figure 1F). This observation may support the findings that AR plays a role in telomere stability and in the replication of telomere DNA in PCa cells [29,30]. Moreover, the upregulation of telomere protecting factors, the expression of the catalytic subunit of telomerase, *hTERT,* is repressed by SAL (Figure 1G). This suggests that androgen treatment at SAL regulates the expression of telomeric genes.

### 3.2. Repression of hTERT by SAL in PCa Tumor Samples Ex Vivo and in Spheroids

To verify the androgen-mediated repression of *hTERT* in PCa tissues, native human PCa specimens from radical prostatectomy were obtained and treated ex vivo for 48 h with or without SAL [8]. Reverse transcription qRT-PCR data confirmed the down-regulation of *hTERT* by SAL in the ex vivo treated patient tumor samples (Figure 1H). This indicates that the AR transrepresses the *hTERT* expression at SAL.

To further verify that androgens repress *hTERT* expression in a 3D tumor cell entity, PCa spheroids from both LNCaP and C4-2 cell lines were generated (Figure 2A,C). 3D-spheroids are suggested to mimic a tumor better in terms of complexity and drug delivery compared with monolayer cultures [24,31,32]. The spheroids were treated with DHT at SAL or R1881 at LAL or SAL, and the growth was compared with the solvent control for the indicated days. Treatment of spheroids with the androgens DHT or R1881 at SAL conditions significantly inhibited the growth of the LNCaP (Figure 2A,B) and C4-2 (Figure 2C,D) spheroids. The SAL-mediated growth repression was slightly more pronounced in LNCaP compared with the C4-2 spheroids. Although LAL promotes the growth of LNCaP cells in adherent cell cultures [8], LAL treatment did not show a significant effect on the growth of LNCaP spheroids. The growth of spheroids CRPC C4-2 cells was unaffected by LAL treatment.

To confirm growth reduction of spheroids by SAL treatment, LNCaP (Figure 2E,F) and C4-2 (Figure 2G,H) were sliced. Immunostaining of the proliferation marker Ki-67 was performed and co-stained with DAPI. Ki-67 staining was found in the outer layer of both the LNCaP and C4-2 spheroids (Figure 2E,G). The data further suggest that SAL treatment strongly reduces Ki-67 staining, indicating that the level of proliferating cells is reduced by SAL in both LNCaP and C4-2 spheroids.

To analyze whether SAL induces cellular senescence, the spheroid slices were analyzed for the senescence-associated β-galactosidase (SA β-Gal) activity (Figure 2F,H). We observed a robust induction of SA β-Gal activity by SAL treatment in the spheroids of LNCaP, which was less pronounced in those derived from C4-2 cells. An increase of SA β-Gal activity was not observed with the LAL treatment. These data suggest that PCa spheroids respond to androgen treatment and that high doses of androgens inhibit growth and induce cellular senescence.

To confirm the inhibition of *hTERT* mRNA levels by SAL also in spheroids, the RNA from both the LNCaP and C4-2 spheroids were extracted. The obtained data suggest that the expression of *hTERT* is reduced by SAL treatment in the 3D spheroid model system of both cell lines (Figure 2I,J). Notably, the *hTERT* expression is repressed more potently by SAL in LNCaP cells compared with the castration-resistant C4-2 cells. This confirms the RNA-seq transcriptome analysis and is consistent with the *hTERT* mRNA expression data of the adherent monolayer. While DHT treatment leads to the inhibition of the hTERT expression in LNCaP spheroids, we observed only mild inhibition of *hTERT* in the C4-2 spheroids.

Thus, SAL treatment leads to a repression *hTERT* expression in adherent PCa monolayer cells, in PCa spheroids, and in patient tumor samples.

### 3.3. Androgen-Mediated Repression and Activation of hTERT Expression Is Time- and Concentration-Dependent

To analyze the concentration-dependent androgen regulation of the *hTERT* expression, RT-qPCR was employed. We compared the androgen response of the known positively regulated AR target gene KLK3 in both human PCa cell lines LNCaP (Figure 3A,B) and C4-2 (Figure 3C,D). As expected, by increasing the androgen concentrations, the KLK3 expression was enhanced in both cell lines, with a stronger effect in the LNCaP compared with the C4-2 cells (Figure 3A,C). An upregulated *KLK3* expression remained induced when higher androgen concentrations were used. However, interestingly, the *hTERT* expression was up-regulated at lower androgen levels, whereas it was inhibited at higher levels of androgens (Figure 3B,D). This antithetic up- and down-regulation was observed in both cell lines. Notably, the androgen-mediated induction, as well as the repression of *hTERT,* was less prominent in the castration-resistant C4-2 cells. This bidirectional response of *hTERT* expression depends on the androgen concentrations, which may explain, at least in part, some of the controversial discussions about its regulation by AR. Thus, the data indicate a dual role of AR in influencing the *hTERT* expression dependent on androgen levels.

Next, we analyzed the time-dependency of the androgen response of hTERT mRNA. The results suggest that the level of *hTERT* mRNA was reduced at 48 and 72 h by SAL treatment (Figure 3E). At a protein level, hTERT was slightly repressed at 48 h and more pronounced at 72 h with the SAL treatment (Figure 3F, the uncropped Western blots are shown in Appendix A). Surprisingly, treatment with the second-generation AR antagonist Enz resulted also in a reduction of the hTERT protein level at 72 h (Figure 3F). This observation is consistent with reduced *hTERT* mRNA levels by different AR antagonists in both cell lines (Appendix A). These results indicate that unexpectedly, the AR antagonists Bicalutamide, Enz, and Darolutamide did not fully act in an opposite manner compared with the agonists in order to regulate hTERT.

### 3.4. Identification of a Positive Androgen Response Element in the Distal hTERT Promoter

To obtain additional molecular insights into the dual regulation of *hTERT* by androgens, we used reporter assays in CV1 cells that lack AR, and the related receptors for glucocorticoids, progesterone, and estrogen that may interfere with the assay system. Without the co-expression of AR, no androgen regulation of *hTERT* was observed [9]. Based on chromatin immunoprecipitation (ChIP) experiments, AR is recruited androgen-dependently to both the −4 kb and −0.1 kb regions of the *hTERT* promoter [9] (Appendix A). Therefore, the promoter fragments spanning these regions were analyzed functionally in more detail using promoter fragments fused to a TATA-box-luciferase construct. The *hTERT* promoter activity spanning the entire region from −4 kb to the start codon is repressed by the androgen treatment (Figure 4A,B). Focusing on the distal −4 kb *hTERT*, various *hTERT* fragments were analyzed for their transcriptional activity. Using *hTERT* −3988 to −3276 and smaller fragments, the androgen treatment led to potent up-regulation of the reporters, suggesting a positive ARE (pARE) in the distal promoter region of *hTERT* (Figure 4A,C,D) at −4 kb. Thus, the data indicate the existence of a pARE that was narrowed down to −3985 to −3956 spanning only 29 bp.

### 3.5. Identification of a Negative Androgen Response Element in the Proximal hTERT Promoter

Focusing on the −0.1 kb region to which AR is also recruited in an androgen-dependent manner, various promoter deletions were generated (Figure 5A). In the absence of AR (empty vector—e.v.) no androgen-mediated transrepression was observed (Figure 5B). Co-expressing AR indicates androgen-mediated repression by fragments spanning the proximal promoter region of −120 bp to −58 bp upstream of the transcription start site as the nARE (Figure 5C). The promoter fragment lacking this sequence lacked androgen-mediated transrepression. In addition, the AR mutant AR-T877A mediated transrepression via this nARE (Figure 5D). The AR-T877A mutant mediated resistance to the AR antagonist flutamide, and was expressed in both LNCaP and C4-2 cells. Using the −240 in both orientations, it was suggested that the transrepressive element was located within this fragment, but acted in an orientation-dependent sense manner. Further deletions of the −120 to −58 bp sequence led to loss of transrepression (Figure 5E). These data suggest the minimal fragment for AR, and the androgen-dependent transrepression of *hTERT* was the 63bp spanning negative response element nARE at −120 to −58 bp.

### 3.6. The Tumor Suppressor ING1 and ING2 Mediate AR-Regulated Transrepression of hTERT

To address the underlying mechanism of AR-mediated gene repression, various co-repressors were over-expressed in CV1 cells, including NCoR, SMRT, Alien, CoREST, and TSGA, without showing a response to the *hTERT* proximal promoter activity (Appendix A). However, the over-expression of ING1 reduced the activity of the proximal −120 to −58bp *hTERT* sequence (Appendix A). The overexpression of ING1 may squelch repressive factors, and the data suggest that ING1 is associated with the AR-mediated repression of *hTERT*.

Therefore, we hypothesized that the tumor suppressor ING1 might be recruited to the *hTERT* proximal promoter. In addition, we also analyzed ING2 recruitment, as both ING1 and ING2 were identified as AR co-repressors [18,19]. For this purpose, ChIP experiments were performed with LNCaP cells to detect whether ING1 and ING2 were recruited to the *hTERT* proximal promoter (Figure 6A). The data suggest that compared with the solvent control (DMSO), AR, ING1, and ING2 were recruited to the *hTERT* promoter by SAL treatment of the LNCaP cells (Figure 6A). Interestingly, the other ING family members ING3, ING4, and ING5 did not show an androgen-dependent recruitment (Appendix A). Based on these data, we further hypothesized that the ING1 and ING2 tumor suppressors are involved in the AR-mediated transrepression of *hTERT*.

Therefore, to confirm and functionally analyze the role of ING1 and ING2 in the *hTERT* expression, knockdown experiments were performed in LNCaP cells using a retroviral expression system. Western blot data indicate a knockdown of ING1 and ING2 (Figure 6B,C, the uncropped Western blots are shown in Appendix A). The vector sh–luc served as a knockdown control. qRT-PCR was performed to analyze the effect on *hTERT* expression upon ING1 or ING2 knockdown. The data suggest that in either ING1 or ING2 knockdown cells, the androgen-mediated *hTERT* repression is alleviated (Figure 6D,E). This suggests that AR-regulated transrepression is mediated by ING1 and ING2. Hence, the data confirm our hypothesis that the tumor suppressors and AR co-repressors ING1 and ING2 functionally mediate the transrepression of androgen-activated AR on *hTERT*. * *p* ≤ 0.05.

## 4. Discussion

Pieces of evidence suggest that supraphysiological androgens inhibit PCa growth. In line with this, pharmacological SAL is currently being used in clinical trials within the BAT to treat metastatic CRPC patients. BAT includes ADT with intermittent cycles of ADT in combination with SAL [3,4]. It is suggested that BAT might prevent adaptive changes in AR expression, leading to a prolonged therapy response of patients [6]. Here, we show that SAL treatment of human PCa cells leads to the repression of *hTERT* expression. The repression of *hTERT* by androgen was observed with the treatment of 48 h in PCa tumor samples ex vivo, as well as in spheroids. Telomerase function is, on the one hand, to protect telomere erosion by DNA replication, but also has a non-telomeric function in promoting cell growth and mediating more independence towards growth factors in tumorigenesis [33,34]. Inhibition of the *hTERT* activity in tumor cells might therefore be beneficial in tumor growth suppression. In the long term, this might be one beneficial aspect of SAL treatment in BAT, as shown by the SAL-suppressed growth of PCa spheroids. Moreover, it is interesting that androgens regulate *hTERT* expression at least in hormone responsive cancers, which might be an important point as an interesting option of drug therapy.

Interestingly, the androgen-mediated transrepression of *hTERT* is androgen-concentration, AR-dependent, and treatment-time dependent. Interestingly, on the one hand, lower androgen concentrations enhance *hTERT* expression. On the other hand, higher doses of androgens at supraphysiological, pharmacological concentrations repress *hTERT* expression. This opposite regulation of the *hTERT* response dependent on the androgen concentration may account for the discrepancies in the findings of how *hTERT* is regulated by androgens. Activation of the *hTERT* expression by androgens was observed [12,13], whereas repression of *hTERT* by androgen was also shown [9,15,17]. The androgen-mediated up-regulation of *hTERT* at LAL may be mediated by the −4 kb pARE, whereas at SAL, the nARE might be overriding the pARE, leading in sum to a SAL-dependent *hTERT* repression. Interestingly, the identified nARE sequence has a very high GC content and overlaps the recently identified putative quadruplex forming sequence in the *hTERT* promoter [35].

In addition, the treatment time of SAL also accounts for the repression. Androgen treatment of one day or less did not lead to an inhibition of *hTERT* mRNA levels, which is in line with previous findings [9]. Longer treatment periods revealed a downregulation of both the *hTERT* mRNA and *hTERT* protein by SAL.

However, additional factors may also account for the discrepancies in the androgen regulation of *hTERT* expression. It may play an important role whether synthetic androgens or the metabolizable DHT were used for experiments. The metabolites of DHT activate the estrogen receptor beta [27]. It is known that the *hTERT* promoter recruits the estrogen receptor, leading to the activation of the *hTERT* expression [36,37]. Thus, in cells with a higher DHT metabolism, there might be a balance between the repression of *hTERT* by DHT and activation by DHT metabolites through the estrogen receptor. A further point that may lead to the different observations is that androgens are produced in PCa cells [38]. Thus, dependent on the particular PCa tumor or PCa subline, the androgen level could be different and contribute specifically to the regulation of the *hTERT* expression. Previously, we reported that the wild-type but not AR mutant T877A mediates the repression of *hTERT* mRNA levels [9]. This effect was observed through analysis within 24 h of the hormone treatment. It is nevertheless possible that the AR-T877A mutant mediates a different response on *hTERT* as the wild-type AR observed. This might also be reflected by the *hTERT* deletion constructs −1301 to −238 and −58 to −1. Both non-overlapping constructs are enhanced more potently by the AR-T877A mutant compared with wild-type AR, and may therefore account for a difference between wild-type and mutant AR.

Notably, the down-regulation of *hTERT* by androgens is not necessarily coupled to an up-regulation of *hTERT* by AR antagonists. In fact, as LAL up-regulates the hTERT expression, we expect a down-regulation from treatment with AR antagonists. As AR antagonists are shown to repress *hTERT*, the conclusion was drawn that androgens would show the opposite activity and may activate *hTERT* expression. However, this seemed to not be the case. We tested various AR antagonists and indeed observed a down-regulation of *hTERT* expression in the same experimental setup using both cell lines (Appendix A), being in line with previous observations [12,39]. The underlying mechanism for antagonist-mediated *hTERT* repression is, however, unclear. Interestingly, the sexual hormones estrogens and androgens regulate the *hTERT* expression at least in hormone responsive cancers, which might be an interesting option for drug therapy.

Thus, the data suggest that when analyzing the regulation of *hTERT* expression, it is critical whether DHT or other synthetic agonists are used, as well as the androgen concentration and the treatment period. In addition, AR antagonists do not act in the opposite manner as agonists when analyzing the *hTERT* expression. These particularities seem to be important factors in analyzing the hormonal regulation of *hTERT*.

## 5. Conclusions

Knowledge about negative hormone response elements, including nAREs, is very limited. The identification of ING1 and ING2 tumor suppressors in the AR-mediated transrepression of *hTERT* provides some first insights into the link between tumor suppressors and the AR (graphical abstract). ING1 and ING2 belong to the ING family of tumor suppressors, with five family members. Both ING1 and ING2 are known as transcriptional repressors interacting with the mSIN3A/HDAC complex [40]. In addition, ING1 and ING2 were shown to act as AR co-repressors that act on the androgen-activated AR [18,19]. This study reveals that the recruitment of both ING1 and ING2 to the proximal promoter of *hTERT* is enhanced by androgen treatment. Moreover, knockdown experiments confirm functionally that ING1 and ING2 are mediated with SAL-induced *hTERT* repression. Thus, the androgen-dependent recruitment of these ING factors could mechanistically explain the androgen-dependent repression of *hTERT*. Taken together, here, we show an antithetic response of *hTERT* expression dependent on the androgen concentrations and the interplay of the ING1 and ING2 tumor suppressors on *hTERT,* indicating a novel pathway.

## Figures and Tables

**Figure 1 cancers-13-04025-f001:**
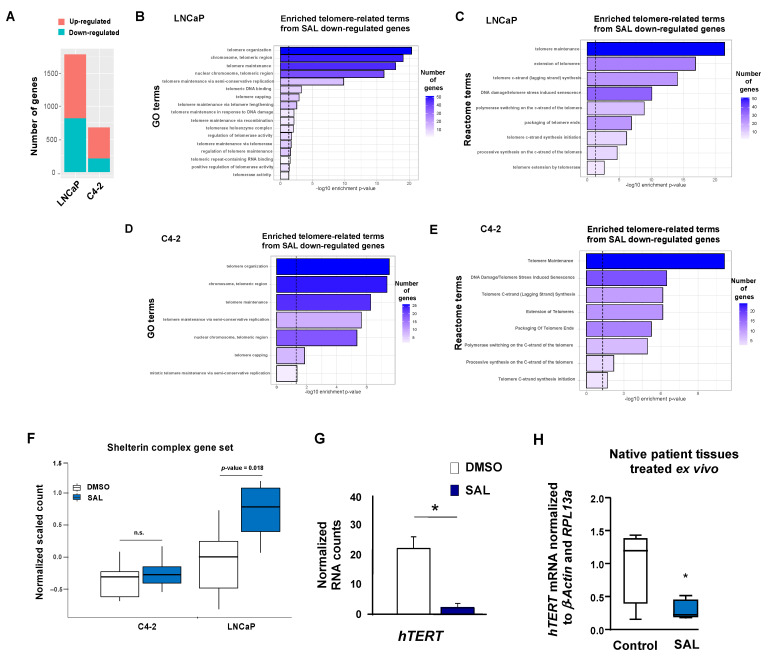
Transcriptome analyses reveal a significant enrichment of telomeric Gene Ontology, Reactome terms, and shelterin complex subunits. The androgen-sensitive LNCaP or the castration-resistant C4-2 human PCa cells were treated with SAL (1 nM R1881) or solvent treated control (DMSO) 72 h prior to the isolation of RNA and performing RNA-seq. Transcriptome data were analyzed bioinformatically comparing SAL treated with solvent control-treated cells. (**A**) Number of differentially expressed genes (DEGs) including the number of upregulated and downregulated genes in both LNCaP and C4-2 cells. The telomere-related downregulated genes exhibit a stronger response to androgen treatment in LNCaP cells considering both the number of DEGs and the intensity of change (log2 fold). Indicated are SAL-mediated down-regulated genes significantly enriched in telomere-related GO and Reactome terms in LNCaP (**B**,**C**) and C4-2 (**D**,**E**) cells. (**F**) Gene set enrichment analysis (GSEA) of telomere-related genes, the subunits of the shelterin complex, upon SAL treatment compared with DMSO. Error-bars represent SEM. Wilcoxon paired test (gene-wise comparison, two-tailed) is used for *p*-value calculation. (**G**) mRNA expression of *hTERT* in LNCaP cells treated with the solvent control DMSO or SAL. Normalized RNA counts are indicated with the values for DMSO as the control. Adjusted *p*-value (* *p* < 0.05) was calculated using the DeSEQ2. (**H**) Native PCa tissue pieces obtained from patients with radical prostatectomy were treated ex vivo with R1881 at SAL as described previously (8). RT-qPCR was used to detect the regulation of *hTERT* expression by SAL. *n* = 4, one-tailed student *t*-test, * *p*-value: 0.028, n.s.: not significant.

**Figure 2 cancers-13-04025-f002:**
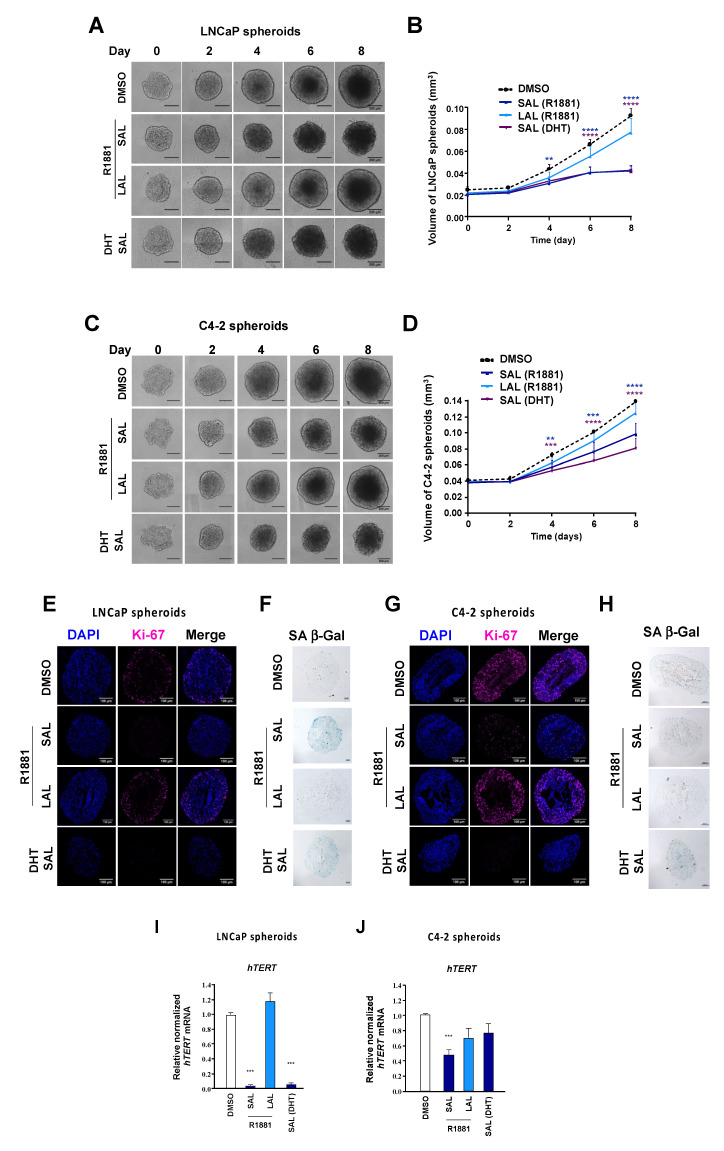
Inhibition of LNCaP or C4-2 spheroid growth and *hTERT* expression by SAL treatment. Spheroids from both LNCaP (**A**,**B**) and C4-2 cells (**C**,**D**) were generated and treated with the indicated ligands 1nM R1881 (SAL), 1pM R1881 (LAL), 10nM DHT (DHT SAL), or the DMSO control for 8 days. The representative pictures of the spheroids from LNCaP (**A**,**B**) and C4-2 (**B**,**D**) were analyzed every second day. The mean ± SEM values were calculated from three independent experiments, whereas for each biological experiment, six technical replicates were performed. The significance of the data from the treatments versus DMSO was analyzed with two-way repeated measure ANOVA with Dunnett’s multiple comparisons test (** *p* ≤ 0.01, *** *p* ≤ 0.001, **** *p* ≤ 0.0001). Both LNCaP (**E**,**F**) and C4-2 (**G**,**H**) spheroids were cultured for 8 days with the indicated ligands, and were sliced (4 µM thick) and subsequently immunostained for Ki-67 as a proliferation marker (**E**,**G**), and finally were analyzed for SA β-Gal activity as a hallmark for senescent cells (**F**,**H**). The scale bar is indicated for 100 µm. Spheroids generated from LNCaP (**I**) or C4-2 (**J**) cells were analyzed for *hTERT* mRNA by qRT-PCR. Values were calculated relative to DMSO, which was set arbitrarily as 1. Error bars indicate SEM. *** *p* ≤ 0.001.

**Figure 3 cancers-13-04025-f003:**
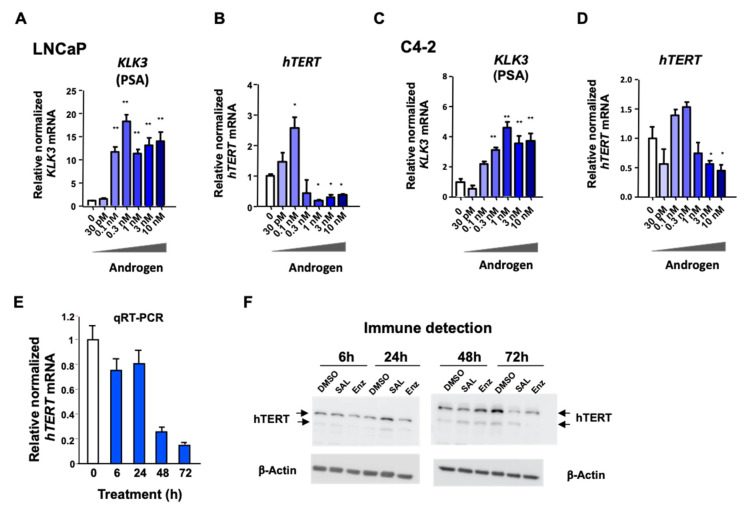
Concentration- and time-dependent response of *hTERT* expression by androgens. LNCaP (**A**,**B**) or C4-2 (**C**,**D**) cells were treated with the indicated concentrations of R1881 for 3 days and were analyzed by qRT-PCR for KLK3 mRNA, encoding PSA, as a positive androgen-responsive gene (**A**,**C**), or for the *hTERT* mRNA levels (**B**,**D**). β-Actin was used for normalization. The values were calculated relative to DMSO, which was set arbitrarily as 1. Error bars indicate SEM. (**E**) *hTERT* expression was analyzed at an mRNA level by qRT-PCR of LNCaP cells treated for the indicated time periods with 1nM R1881. Indicated are the relative values normalized to house-keeping gene β-Actin and RPL13a. (**F**) Western blotting indicates the protein level of the two hTERT isoforms (arrows) at the indicated treatment times. The treatment with second-generation AR antagonist Enzalutamide (Enz; 10 µM) was included. β-Actin was used for normalization. DMSO served as the solvent control. *p*-values: * *p* ≤ 0.05, ** *p* ≤ 0.01.

**Figure 4 cancers-13-04025-f004:**
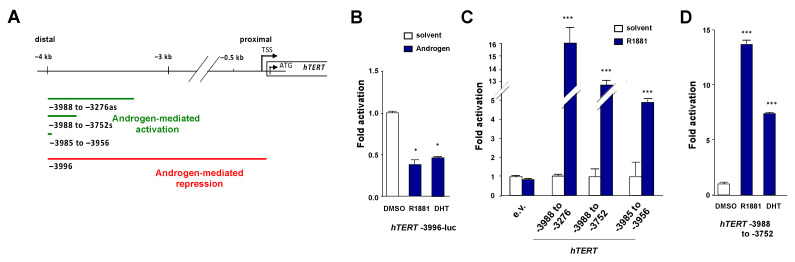
Identification of a positive androgen response element in the hTERT distal promoter. CV1 cells lacking endogenous functional AR and other steroid receptors were co-transfected with AR and pCMV lacZ. The obtained values were normalized to β-galactosidase activity after being co-transfected pCMV-lacZ. e.v.—empty vector; R1881 (10 nM); DHT (10 nM). (**A**) Overview of the *hTERT* deletion constructs used for the reporter assays. The sections highlighted in green or red color represent the promoter construct activities that mediate an enhanced or repressed reporter activity by androgen treatment, respectively. s—sense; as—antisense. (**B**–**D**) Luciferase assays with the indicated *hTERT* promoter regions spanning the −4 kb to −1 kb and distal promoter region constructs. * *p* ≤ 0.05, *** *p* ≤ 0.001.

**Figure 5 cancers-13-04025-f005:**
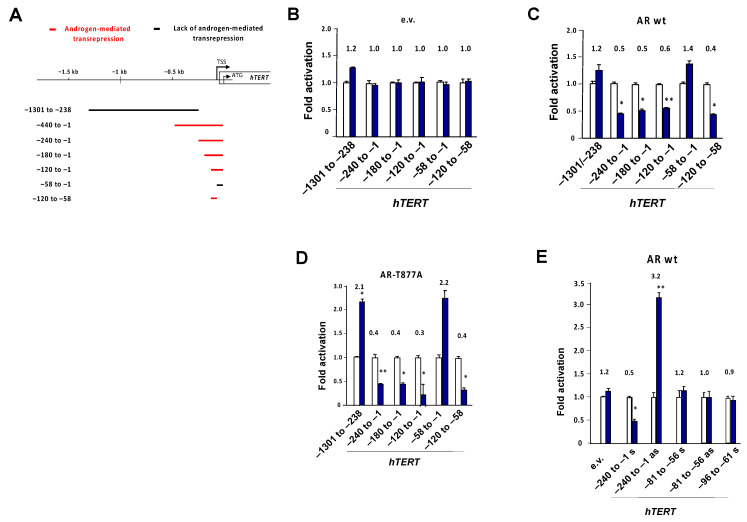
Identification of a negative androgen response element in the *hTERT* proximal promoter. (**A**) Overview of proximal *hTERT* deletion constructs for the reporter assays used. The sections highlighted in red are indicative of androgen-mediated repression, whereas the black line does not suggest an androgen-mediated response. (**B**–**E**) Luciferase assays with indicated *hTERT* promoter spanning the proximal promoter region constructs in similar experimental setup as in Figure 4. e.v.—empty vector lacking *hTERT* sequences or (**E**) the cDNA for the receptor (**B**); ARwt—wild-type AR; AR-T877A—mutant AR, R1881 (1 nM). Numbers above the bars indicate the fold induction by androgen treatment. s—sense; as—antisense orientation of the inserted *hTERT* fragment. * *p* ≤ 0.05, ** *p* ≤ 0.01.

**Figure 6 cancers-13-04025-f006:**
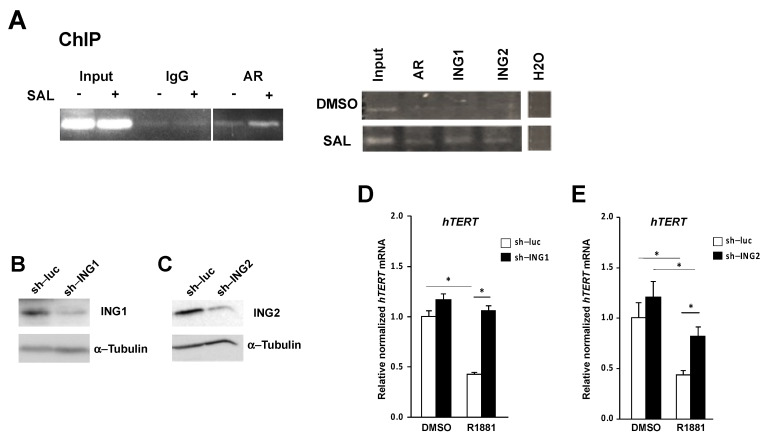
The tumor suppressors ING1 and ING2 mediate the androgen-regulated transrepression of *hTERT. (***A**) ChIP experiments using antibodies against AR, ING1, and ING2 for LNCaP cells treated with either DMSO as the solvent control or SAL. Immunoprecipitates were analyzed by nested PCR for the proximal *hTERT* region. (**B**,**C**) Western blot of ING1 (**B**) or ING2 (**C**) of LNCaP cell extracts using shRNA knockdown for ING1 (sh-ING1), ING2 (sh-ING2), or the control vector (sh-luc). α-Tubulin detection served as the loading control. (**D**,**E**) qRT-PCR of endogenous *hTERT* expression with and without ING1 knockdown (**D**) or ING2 (**E**) knockdown in the presence of androgen or DMSO as the solvent control. * *p* ≤ 0.05.

**Table 1 cancers-13-04025-t001:** Antibodies used for ChIP experiments.

Primary Antibody	Manufacturer/Catalog No.	Dilution
Anti-androgen receptor, polyclonal, and rabbit	Merck, 06-680	1:15
Anti-ING1, polyclonal, rabbit	Sigma Aldrich, SAB1410716-100UG	1:20
Anti-ING2, polyclonal, rabbit	Proteintech, 11560-1-AP	1:10
Anti-rabbit IgG, HRP-conjugated	Cell Signaling, 70,745	1:100

## Data Availability

The RNA-sequencing data are available in the gene expression omnibus (GEO) database under the accession number GSE151492.

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
