# Peer review of "Antithetic hTERT Regulation by Androgens in Prostate Cancer Cells: hTERT Inhibition Is Mediated by the ING1 and ING2 Tumor Suppressors"

_cancers, 2021, doi:10.3390/cancers13164025_

Round 1
Reviewer 1 Report
This manuscript describes that we reveal that androgens repress and activate hTERT expression in a concentration-dependent manner. The authors describe supraphysiological androgen levels (SAL)-mediated gene repression of hTERT in prostate cancer (PCa) cell lines, human PCa samples from patients treated ex vivo and in cancer spheroids derived from androgen-dependent or castration resistant PCa (CRPC) cells. ChIP with functional assays revealed a positive (pARE) and a negative androgen response element (nARE). The androgen receptor (AR) and the tumor suppressors, inhibitor of growth 1 and 2 (ING1 and ING2), are androgen-dependently recruited. Mechanistically, knockdown indicate that ING1 and ING2 mediate AR-regulated transrepression.
Characteristics of mutant AR-T877A should be described.
What is “control” in Fig. 5E?
How does an antisense sequence have an opposite effect in Fig. 5E? Citations are required.
How did you select “-81” and “-96” in Fig. 5E?
Fig. 6A requires negative control. It is recommended to add measurements by real-time PCR for this ChIP assays.
Using a slash (/) between 2 numbers (-3988/-3276, for example) does not represent the region of the sequence between the numbers.
Author Response
Reviewer 1
This manuscript describes that we reveal that androgens repress and activate hTERT expression in a concentration-dependent manner. The authors describe supraphysiological androgen levels (SAL)-mediated gene repression of hTERT in prostate cancer (PCa) cell lines, human PCa samples from patients treated ex vivo and in cancer spheroids derived from androgendependent or castration resistant PCa (CRPC) cells. ChIP with functional assays revealed a positive (pARE) and a negative androgen response element (nARE). The androgen receptor (AR) and the tumor suppressors, inhibitor of growth 1 and 2 (ING1 and ING2), are androgendependently recruited. Mechanistically, knockdown indicate that ING1 and ING2 mediate ARregulated transrepression.
1. Characteristics of mutant AR-T877A should be described.
Thank you for pointing this out. We have added this information now on page 13.
2. What is “control” in Fig. 5E?
The control is the empty reporter vector lacking hTERT sequences. We have indicated that now in the revised Fig. 5 legend.
3. How does an antisense sequence have an opposite effect in Fig. 5E? Citations are required.
That is an interesting point. We believe that the transrepressive activity is located on the fragment -240 but acts in an orientation-dependent manner. A transactivation is mediated presumably by another element in case moving the -240 position close to the TATA box.
Indeed, the transrepressive activity was further narrowed down to -120.
We have now addressed this observation on page 13.
4. How did you select “-81” and “-96” in Fig. 5E?
Yes, after having not detected transrepression by the -81 fragment we generated and analyzed the -96 promoter fragment.
- Fig. 6A requires negative control. It is recommended to add measurements by real-time PCR for this ChIP assays.
We have added now the IgG control. See revised Fig. 6.
6. Using a slash (/) between 2 numbers (-3988/-3276, for example) does not represent the region of the sequence between the numbers.
We have changed the labeling avoiding a slash. See revised figures 4 and 5.
We believe the manuscript has been improved and would like to thank for your
consideration.
Sincerely,
Reviewer 2 Report
In the current study, authors were trying to demonstrate that AR could repress hTERT gene by supraphysiological level of androgen (SAL) through recruitment of both ING1 and ING2 to the proximal promoter of hTERT. Moreover, authors demonstrated knocking down of ING1 and ING2 diminished SAL-induced hTERT repression.
- In Fig1, authors should describe in the main text that what the concentration of R1881 at supraphysiological androgen level (SAL) that treated the LNCaP and C4-2 was and performed RNA-Seq with? Can authors summarize the RNA-seq results in general by showing the number of differentially expressed genes (DEGs), including the number of upregulated genes and number of dowregulated genes with a log2 fold change of 1. Also, please show the list of DEGs.
- In Fig1, 72 hours of R1881 treatment is too long for RNA-seq, the DEGs may be indirectly regulated by R1881. Please validate your RNA-seq results by treating LNCaP and C4-2 cells with DHT at same concentration as R1881 used in RNA-seq for 24 hours, test regulations for some top DEGs generated from the RNA-seq results.
- In Fig1, Also, authors should show the overall Gene Ontology (GO term) results from the RNA-seq to validate the sequencing results.
- In Fig1F, please find some genes from the shelterin complex gene set to validate the result by showing the mRNA expression induced by the same concentration of DHT.
- In Fig1G, is this the mRNA expression of hTERT treated with SAL concentration of R1881? If it is, the hTERT mRNA expression should be relative to the GAPDH or other housekeeping gene expression in the same reaction.
- Fig3B,C,D are completely blocked by other figure in the uploaded version of the manuscript for peer review.
- Fig 3F is also hard to see, what are the two blots? What are the differences? Why the arrow pointed hTERT expression are different in size? One of the b-actin expression is very weak, please generate a new set of data.
- Fig4A, please use any Genome Browser to show the AR binding +/- DHT/R1881 at SAL level in LNCaP to show the binding site of AR at the proximal and distal enhancer sites.
- Please describe Reporter Gene Assay in more details.
- When describing transcription repression activities of AR, it is due to the recruitment of AR co-repressor to exert the transcription repression activity. In Fig 4B and 5B-E, How transfection of fragment of DNA (hTERT-3996) into a AR-negative cell and shows a lower luciferase activity than the baseline, which is no luciferase activity?
- Fig5A analysis needs to be presented with AR binding in Genome Browser.
- Fig6A, ChIP AR, ING1 and ING2 analyses need to compare with IgG, the same species of the specific antibodies, to show it is more enriched than unspecific binding.
Author Response
Reviewer 2
In the current study, authors were trying to demonstrate that AR could repress hTERT gene by supraphysiological level of androgen (SAL) through recruitment of both ING1 and ING2 to the proximal promoter of hTERT. Moreover, authors demonstrated knocking down of ING1 and ING2 diminished SAL-induced hTERT repression.
1. In Fig1, authors should describe in the main text that what the concentration of R1881 at supraphysiological androgen level (SAL) that treated the LNCaP and C4-2 was and performed RNA-Seq with? Can authors summarize the RNA-seq results in general by showing the number of differentially expressed genes (DEGs), including the number of upregulated genes and number of dowregulated genes with a log2 fold change of 1. Also, please show the list of DEGs.
- We have added now the information for androgen concentration also in the main text. See page 7.
- We added the DEG numbers of significantly up- and downregulated genes with a log2 fold change of 1. See revised Fig. 1A
- In addition, we provide the data with the DEG genes. See supplementary DEG excelfile.
2. In Fig1, 72 hours of R1881 treatment is too long for RNA-seq, the DEGs may be indirectly regulated by R1881. Please validate your RNA-seq results by treating LNCaP and C4-2 cells with DHT at same concentration as R1881 used in RNA-seq for 24 hours, test regulations for some top DEGs generated from the RNA-seq results.
- The purpose was to analyze longer treatments by androgens, especially at SAL as it is be applied in the bipolar androgen therapy. Shorter treatments have a big disadvantage that detection of transrepression relies on the mRNA stability / degradation.
3. In Fig1, Also, authors should show the overall Gene Ontology (GO term) results from the RNA-seq to validate the sequencing results.
- We added the excel file with the GO terms for both cell lines. See both supplementary GO-term excel- files.
- In Fig1F, please find some genes from the shelterin complex gene set to validate the result by showing the mRNA expression induced by the same concentration of DHT.
- We found that some shelterin complex subunit encoding genes are not and some are upregulated. We have added the data in supplement. However, since DHT can be metabolized into estrogen receptor agonist (Handa et al., 2008) and since it is known that estrogens may counteract androgen regulation this experiment will not allow to make a statement about androgen specific effect. This is mentioned on page 7.
5.In Fig1G, is this the mRNA expression of hTERT treated with SAL concentration of R1881? If it is, the hTERT mRNA expression should be relative to the GAPDH or other housekeeping gene expression in the same reaction.
- The data presented in Fig. 1G derive from RNA-seq data with the identical treatments using the AR specific ligand R1881. In RNA-seq, the number of reads for each gene is normalized to the total number of sequenced reads which is much more precise than normalizing to only one house-keeping gene (HKG). Normalizing to the HKG is usually used for qRT-PCR and it is not recommended for the RNA-seq. Nevertheless, we show in Fig. 2 and 3 the qRT-PCR data normalized to HKG and confirm the RNA-seq data.
6. Fig3B,C,D are completely blocked by other figure in the uploaded version of the manuscript for peer review.
- This is surprising since we have looked at the word and PDF files prior submission and did not see an overlap. Sorry for the inconvenience. We revised the figures.
7. Fig 3F is also hard to see, what are the two blots? What are the differences? Why the arrow pointed hTERT expression are different in size? One of the b-actin expression is very weak, please generate a new set of data.
- The two blots show the time-dependent expression of both hTERT isoforms (120 and 127kDa) at protein level. We have now revised the figure.
8. Fig4A, please use any Genome Browser to show the AR binding +/- DHT/R1881 at SAL level in LNCaP to show the binding site of AR at the proximal and distal enhancer sites.
- Fig. 4A is meant to show schematically the promoter deletion constructs used in Fig. 4.
We show now the AR binding sites in the genome browser. See new supplemental Fig. S2. - Please describe Reporter Gene Assay in more details.
- The reporter assay is a standard well-established assay used since decades. We
added some additional information in material method section.
10. When describing transcription repression activities of AR, it is due to the recruitment of AR co-repressor to exert the transcription repression activity. In Fig 4B and 5B-E, How transfection of fragment of DNA (hTERT-3996) into a AR-negative cell and shows a lower luciferase activity than the baseline, which is no luciferase activity?
- Each promoter has a basal activity this includes also hTERT. Androgen treatment represses this basal activity, which is central to this manuscript. Perhaps there is a misunderstanding since the hTERT-3996 is fused to the luciferase gene. Similarly, all the promoter deletions are inserted in front of a TATA-box-luciferase fusion. We emphasized this point in Material methods section.
11. Fig5A analysis needs to be presented with AR binding in Genome Browser.
- Fig. 5A is meant to show schematically the promoter deletion constructs used in Fig.5.
We show now the AR binding sites in the genome browser. See supplemental Fig. S2.
12. Fig6A, ChIP AR, ING1 and ING2 analyses need to compare with IgG, the same
species of the specific antibodies, to show it is more enriched than unspecific binding.
- We have now added the IgG control. See revised Fig. 6A.
We confirm that neither the submitted paper nor a similar paper is currently submitted to
another journal.
We believe the manuscript has been improved and would like to thank for your
consideration.
Sincerely,
Round 2
Reviewer 2 Report
I absolutely appreciate how much work authors have put into this project. The major point of the androgen-mediated TERT repression is AR binding at TERT upon androgen induction and recruitment of co-repressors, in Fig4-6. From published AR ChIP-Seq studies, 90% of the AR binding sites are located at distal enhancer regions.
My question about showing AR binding at author identified TERT promoter sites in LNCaP or other prostate cancer (PCa) tumor samples is the basis of the study. The AR binding peaks what I hopefully to see is in the attached, not as shown in Fig S2 by the authors that only shows TERT reference gene transcripts. There might be a small AR binding peak at the TERT promoter site from one PCa tumor sample shown in the attached figure. In Fig4 and 5, I appreciate that authors were studying the AR activities at specific TERT promoter regions, but, with AR binding peaks will be easy for the readers.
Overall I believe this is quite interesting study, hope the field can benefit from it.

Author Response
Thank you for pointing this out.
There may be a misunderstanding. AR binding peaks are obtained by ChIP-seq. We have performed ChIP-qPCR specifically focusing on the AR binding site that was previously known and that coincides with the responsive promoter site, which mediates androgen-induced transrepression, a very rare analyzed phenomenon.
I agree showing peaks is also interesting. We are aware of distal binding sites of AR on the hTERT region identified by ChIP-seq. However, the proximal promoter core promoter has an extremely high accumulation of GC sequences that are not easily picked up by ChIP-seq detection. That is perhaps the reason why particular this proximal hTERT promoter region forms three parallel DNA G-quadruplexes (Monsen et al., 2020).
The ChIP-qPCR data show recruitment of AR in an androgen-dependent manner on the proximal hTERT core promoter, similar to that of the AR corepressors and tumor suppressors ING1 and ING2. Alongside with ING1 and ING2 knockdown, these data were provided to reveal the novel molecular mechanism of transrepression by AR on hTERT, a key factor of tumorigenesis.